# HARMONY IN DIVERSITY: MERGING NEURAL NETWORKS WITH CANONICAL CORRELATION ANALYSIS

**Stefan Horoi** [1,3]    **Albert Manuel Orozco Camacho** [2,3]    **Eugene Belilovsky** [2,3]    **Guy Wolf** [1,3]

[1]Université de Montréal    [2]Concordia University    [3]Mila - Quebec AI Institute

{stefan.horoi, guy.wolf}@umontreal.ca
{albert.orozcocamacho, eugene.belilovsky}@concordia.ca

## ABSTRACT

Ensembling multiple models enhances predictive performance by utilizing the varied learned features of the different models but incurs significant computational and storage costs. Model fusion, which combines parameters from multiple models into one, aims to mitigate these costs but faces practical challenges due to the complex, non-convex nature of neural network loss landscapes, where learned minima are often separated by high loss barriers. Recent works have explored using permutations to align network features, reducing the loss barrier in parameter space. However, permutations are restrictive since they assume a one-to-one mapping between the different models' neurons exists. We propose a new model merging algorithm, CCA Merge, which is based on Canonical Correlation Analysis and aims to maximize the correlations between linear combinations of the model features. We show that our method of aligning models leads to better performances than past methods when averaging models trained on the same, or differing data splits. We also extend this analysis into the harder many models setting where more than 2 models are merged, and we find that CCA Merge works significantly better in this setting than past methods. [1]

## 1 INTRODUCTION

A key strategy for improving the performance and robustness of machine learning models involves utilizing multiple models which capture diverse, potentially complementary insights from data. Ensembles, which combine outputs from several trained models through averaging or majority vote at inference time, improve predictive performance but at the expense of increased memory and computational costs (Ho, 1995; Lobacheva et al., 2020). An alternative, model fusion, merges parameters from multiple models into one by averaging them, reducing storage and computation costs but facing challenges in maintaining performance (Frankle et al., 2020; Stoica et al., 2024). Indeed, while multiple good local minima can be found for a given model and task, these minima are most often separated by regions of high loss (Frankle et al., 2020). Combining the parameters of trained models is likely to lead to a high loss region in parameter space, thus destroying the learned features.

*Linear mode connectivity* which describes two optima connected by a *linear* low loss path in the parameter space (Frankle et al., 2020), provides a straightforward way of combining models through linear interpolation of their parameters, the existing low loss path guaranteeing the preserving of performance. However, LMC is very rare in practice and is not guaranteed even for networks with the same initializations (Frankle et al., 2020), and it is only made rarer by NN's permutation invariance. Entezari et al. (2022) conjectured that most SGD solutions can be permuted in such a way that they are linearly mode connected to most other SGD solutions. Many works in recent years have provided algorithms for finding permutations that successfully achieve LMC between pairs of SGD solutions, or at least significantly lower the loss barrier on the linear path between these solutions, further supporting this conjecture (Singh & Jaggi, 2020; Peña et al., 2023; Ainsworth et al., 2023).

However, there is nothing inherently stopping NNs from distributing computations that are done by one neuron in a model to be done by multiple neurons in the other model. Permutations would fail to capture this relationship since they can only account for one-to-one mappings between features. Furthermore, the focus of recent works has been mainly on merging pairs of models. However, if a

---

[1]Our code is publicly available at `https://github.com/shoroi/align-n-merge`

similar function is learned by networks trained on the same task then it should be possible to extract the commonly learned features from not only two but also a larger population of models.

**Contributions**  Our main contributions are threefold: **(1)** We propose a new model merging method based on Canonical Correlation Analysis (Sec. 3) which we will refer to as "CCA Merge". This method is more flexible than past, permutation-based methods and therefore makes better use of the correlation information between neurons (Sec. A.1). **(2)** We compare our CCA Merge to past works and find that it yields better performing merged models across a variety of architectures and dataset. This is true in both settings where the models were trained on the same data (Sec. 4.1) or on disjoint splits of the data (Sec. 4.3). **(3)** We take on the difficult problem of aligning features from multiple models and then merging them. We find that CCA Merge is significantly better at finding the common learned features from populations of NNs and aligning them, leading to lesser accuracy drops as the number of models being merged increases (Sec. 4.2).

## 2 RELATED WORK

**"Easy" settings for model averaging**  Linear mode connectivity is hard to achieve in deep learning models (Frankle et al., 2020) and it seems that only models which are already very close in parameter space can be directly combined through linear interpolation. This is the case for snapshots of a model taken at different points during its training trajectory (Garipov et al., 2018; Izmailov et al., 2018), multiple fine-tuned models with the same pre-trained initialization as is standard in NLP (Wortsman et al., 2022; Ilharco et al., 2023; Yadav et al., 2023) or models merged every couple of epochs during training as in federated learning settings (McMahan et al., 2017). We emphasize that these settings are different from ours in which we aim to merge *fully trained models* with different parameter initializations and SGD noise (data order and augmentations).

**Merging multiple models**  Merging more than two models has only been explored thoroughly in the "easy" settings stated above (Wortsman et al., 2022; Jolicoeur-Martineau et al., 2023; Yadav et al., 2023). The only works focused on providing feature alignment methods which have also applied these methods to the multiple model merging setting are Ainsworth et al. (2023) with their "Merge-Many" algorithm and Singh & Jaggi (2020). However their results for merging multiple models are mainly relegated to the appendix and focus on easy settings such as MLPs on MNIST or they fine-tune the resulting merged model. We extend this line of work to more challenging settings, using more complex model architectures, we report the merged models accuracies directly without fine-tuning and make this a key focus in our work.

**Model merging beyond permutations**  We note that the two model merging methods based on optimal transport Singh & Jaggi (2020) and Peña et al. (2023) can also align models beyond permutations. However, in Singh & Jaggi (2020) this only happens when the two models being merged have different numbers of neurons at each layer, which is not the case for the majority of their experiments. The method proposed by Peña et al. (2023) isn't constrained to finding binary permutation matrices but binarity is still encouraged through the addition of an entropy regularizer. For a more complete account of related works we direct the reader to the Appendix.

## 3 USING CCA TO MERGE MODELS

**Merging Models**  Consider two deep learning models $\mathcal{A}$ and $\mathcal{B}$ with the same architecture. Let $\{L_i^{\mathcal{M}}\}_{i=1}^N$ be the set of layers of model $\mathcal{M} \in \{\mathcal{A}, \mathcal{B}\}$ and let $X_i^{\mathcal{M}} \in \mathbb{R}^{m \times n_i}$ denote the set of outputs of the $i$-th layer of model $\mathcal{M}$ in response to $m$ given inputs. We assume $X_i^{\mathcal{M}}$ is centered. We are interested in merging the parameters from models $\mathcal{A}$ and $\mathcal{B}$ in a layer-by-layer fashion. In practice, it is often easier to keep model $\mathcal{A}$ fixed and to find a way to transform model $\mathcal{B}$ such that the average of their weights can yield good performance. Mathematically, we are looking for invertible linear transformations $T_i \in \mathbb{R}^{n_i \times n_i}$ which can be applied at the output level of model $\mathcal{B}$ layer $i$ parameters to maximize the "fit" with model $\mathcal{A}$'s parameters and minimize the interpolation error. The inverse transform is then applied at the input level of the next layer to keep the flow of information consistent inside a given model. With these transformations, $\sigma$ being the non-linearity and $W, b$ being the weight and bias parameters, the output of the transformed layer $i$ of $\mathcal{B}$ becomes:

$$x_i^{\mathcal{B}} = \sigma(T_i W_i^{\mathcal{B}} T_{i-1}^{-1} x_{i-1}^{\mathcal{B}} + T_i b_i^{\mathcal{B}})$$

After finding transformations $\{T_i\}_{i=1}$ for every merging layer in the network we can merge the two model's parameters:

$$W_i = \frac{1}{2}(W_i^{\mathcal{A}} + T_i W_i^{\mathcal{B}} T_{i-1}^{-1}) \tag{1}$$

**Canonical Correlation Analysis** Canonical Correlation Analysis (CCA) is a statistical method aiming to find relations between two sets of random variables. Suppose we have two datasets $X$ and $Y$ both of sizes $n \times d$, where $n$ is the number of instances or samples, and $d$ is the dimensionality or the number of features. Further, suppose that these datasets are centered so that each column has a mean of 0. CCA finds vectors $p_X$ and $p_Y$ in $\mathbb{R}^d$ such that the projections $Xp_X$ and $Yp_Y$ have maximal correlation and norm 1. By iteratively finding such vectors, with the added constraint that each new vector must be orthogonal to the other ones, CCA can find the full projection matrices $P_X$ and $P_Y$ aligning $XP_X$ and $YP_Y$. For more details we direct the reader to De Bie et al. (2005).

**CCA Merge: Merging models with CCA** To merge models using CCA we simply apply the CCA algorithm to the two sets of activations $X_i^{\mathcal{A}}$ and $X_i^{\mathcal{B}}$ to get the corresponding projection matrices $P_i^{\mathcal{A}}$ and $P_i^{\mathcal{B}}$. Using the framework described above of matching model $\mathcal{B}$ to model $\mathcal{A}$, which is consistent with past works, we can define $T_i = \left( P_i^{\mathcal{B}} P_i^{\mathcal{A}^{-1}} \right)^{\top}$. The transpose here is to account for the fact that $T_i$ multiplies $W_i$ on the left while the $P_i$s were described as multiplying $X_i$ on the right. This transformation can be thought of as first bringing the activations of model $\mathcal{B}$ into the common, maximally correlated space between the two models by multiplying by $P_i^{\mathcal{B}}$ and then applying the inverse of $P_i^{\mathcal{A}}$ to go from the common space to the embedding space of $\mathcal{A}$. The averaging of the parameters of model $\mathcal{A}$ and transformed $\mathcal{B}$ can then be conducted following Eq. 1.

# 4 RESULTS

## 4.1 MODELS MERGED WITH CCA MERGE ACHIEVE BETTER PERFORMANCE

In Table 1 we show the test accuracies of merged VGG11 (Simonyan & Zisserman, 2015), ResNet20 and ResNet18 (He et al., 2016) models of different widths trained on CIFAR10 (Krizhevsky & Hinton, 2009), CIFAR100 and ImageNet200 (Russakovsky et al., 2015) (full-sized images but training and evaluating only on 200 of the 1k classes) for CCA Merge and multiple other popular model merging methods. The models were trained either using the one-hot encodings of the labels or the CLIP (Radford et al., 2021) embeddings of the class names as training objectives. We report the average accuracies of the base models being merged under the label "Base net avg." (i.e. each model is evaluated individually and their accuracies are then averaged) as well as the accuracies of ensembling the models (the logits of the different models are averaged and the final prediction is the argmax). Ensembling is considered to be the upper limit of what model fusion methods can achieve. Also, since the models being merged were trained on the same data, we do not expect the merged models to outperforms the base ones in this particular setting.

We compare CCA Merge with the following methods: **Direct averaging:** averaging the models' weights without any neuron alignment; **Permute:** permuting model weights to align them, the permutation matrix is found by solving the linear sum assignment problem consisting of maximizing the sum of correlations between matched neurons (Li et al., 2015; Tatro et al., 2020); **Matching Weights:** main method from Ainsworth et al. (2023); **ZipIt!:** model merging method proposed by Stoica et al. (2024). For ResNets, we recompute the BatchNorm statistics after the weight averaging and before evaluation as suggested by Jordan et al. (2023) to avoid variance collapse.

VGG11 models merged with CCA Merge have significantly higher accuracies than models merged with any other method, and this is true across all model widths considered. Differences in accuracy ranging from 10% ($\times 8$ width) up to 25% ($\times 1$ width) can be observed between CCA Merge and the second-best performing method. Furthermore, CCA Merge is more robust when merging smaller width models, incurring smaller accuracy drops than other methods when the width is decreased from $\times 8$ to $\times 1$; 1.71% drop for CCA Merge versus 18.34% for Matching Weights and 8.19% for Permute. Lastly, CCA Merge seems to be more stable across different initializations, the accuracies having smaller standard deviations than all other methods for the same width except for Matching Weights for $\times 8$ width models. We note that for VGG models with width multipliers above $\times 2$ and ResNet18 on ImageNet200, we ran into out-of-memory issues when running ZipIt!, which is why those results are not present. The same conclusions seem to hold for ResNet20 trained on CIFAR100 and ResNet18 on ImageNet200, although the differences in performance here are less pronounced. For ImageNet, the top 5 accuracy of models merged with CCA Merge is remarkably close to the accuracy of model ensembles, the peak of attainable performance.

| Method | VGG11 on CIFAR10 $\times 1$ | $\times 8$ | ResNet20$\times$8 CIFAR100 | ResNet18$\times$4 on ImageNet200 Top 1 Acc. | Top 5 Acc. |
|---|---|---|---|---|---|
| Base net avg. | 87.27$\pm$0.25 | 88.20$\pm$0.45 | 78.77$\pm$0.28 | 82.09$\pm$0.13 | 95.21$\pm$0.09 |
| Ensemble | 89.65$\pm$0.13 | 90.21$\pm$0.24 | 80.98$\pm$0.21 | 83.51$\pm$0.01 | 95.9$\pm$0.03 |
| Direct averaging | 10.54$\pm$0.93 | 10.45$\pm$0.74 | 14.00$\pm$1.66 | 2.94$\pm$0.12 | 10.52$\pm$0.60 |
| Permute | 54.39$\pm$6.45 | 62.58$\pm$3.31 | 72.90$\pm$0.08 | 71.84$\pm$0.53 | 91.40$\pm$0.24 |
| Matching Weights | 55.40$\pm$4.67 | 73.74$\pm$1.77 | 74.29$\pm$0.51 | 69.37$\pm$0.48 | 90.53$\pm$0.24 |
| ZipIt! | 52.93$\pm$6.37 | - | 72.47$\pm$0.41 | - | - |
| **CCA Merge** | **82.65$\pm$0.73** | **84.36$\pm$2.09** | **75.06$\pm$0.18** | **76.38$\pm$0.20** | **93.03$\pm$0.21** |

Table 1: VGG11$\times$1 and $\times$8 trained on CIFAR10, ResNet20$\times$8 trained on CIFAR100 & ResNet18x4 trained on ImageNet200 - Accuracies and standard deviations from 4 different merges (3 for ImageNet200) of 2 models are presented. Models averaged with CCA Merge notably outperform models merged with other methods even on the significantly harder ImageNet200 task, narrowing the gap between merged models and model ensembles. Model ensembles are significantly more memory and compute expensive and represent the upper bound of attainable performance for model merging. For ImageNet top 5 accuracy CCA Merge remarkably approaches the accuracy of model ensembles.

For both VGG and ResNet architectures and for all considered datasets the added flexibility of CCA Merge over permutation-based methods seems to benefit the merged models. Aligning models using linear combinations allows CCA Merge to better model relationships between neurons and to take into account features that are distributed across multiple neurons.

## 4.2 CCA MERGING FINDS BETTER COMMON REPRESENTATIONS BETWEEN MANY MODELS

In this section, we present our results related to the merging of many models, a significantly harder task. This constitutes the natural progression to the problem of merging pairs of models and is a more realistic setting for distributed or federated learning applications where there are often more than 2 models. Furthermore, aligning populations of neural networks brings us one step closer to finding the common learned features that allow different neural networks to perform equally as well on complex tasks despite having different initializations, data orders, and data augmentations.

The authors of Ainsworth et al. (2023) introduced "Merge Many", an adaptation of Matching Weights for merging a set of models. A simpler way of extending any model merging method to the many models setting is to choose one of the models in the group as the *reference model* and to align every other network in the group to it. Then the reference model and all the other aligned models can be merged. It is by using this "all-to-one" merging that we extend CCA Merge, Permute, and Matching Weights to the many model settings. ZipIt! is naturally able to merge multiple models since it aggregates all neurons and merges them until the desired size is obtained.

In Fig. 1 we show the accuracies of the merged models as the number of models being merged increases. For both VGG and ResNet architectures aligning model weights with CCA continues to yield better performing merged models. In fact, models merged with CCA Merge applied in an all-to-one fashion maintain their accuracy relatively well while the other ones see their accuracies drop significantly. In the VGG case, the drop for other methods is drastic, all merged models having less than 20% accuracy when more than 3 models are being merged while CCA Merge suffers a drop in accuracy of less than 3% when going from 2 to 5 models, staying around the 80% mark. We also note that Merge Many performs only slightly better than its 2 model counterpart (Matching Weights applied in an all-to-one fashion). For ResNets, the accuracy of models merged with Permute drops by ~15% when going from merging 2 models to 20. CCA Merge on the other hand is a lot more robust, incurring a less than 4% drop in accuracy even as the number of models is increased to 20.

These results suggest that CCA Merge is significantly better than past methods at finding the "common features" learned by groups of neural networks and aligning them. The limitations of permutation-based methods in taking into account complex relationships between neurons from different models are highlighted in this context. Here it is harder to align features given that there are more of them to consider and therefore easier to destroy the features when averaging them.

## 4.3 CCA MERGE IS BETTER AT COMBINING FEATURES FROM DIFFERENT DATA SPLITS

In this section we consider the more realistic setting where the models are trained on disjoint splits of the data, therefore they're expected to learn (at least some) different features. Such a set-up is

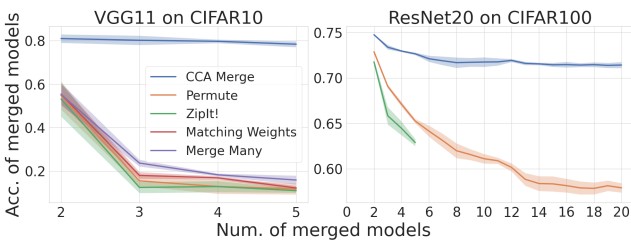

Figure 1: Accuracies of averaging multiple models after feature alignment with different model fusion methods. The mean and standard deviation across 4 random seeds are shown.

| Method | (1) 80%-20% | (2) Dirichlet | (3) 50 classes |
|---|---|---|---|
| Base models avg. | 65.66 ±0.71% | 59.98 ±1.80% | 41.42 ±0.54% |
| Ensemble | 77.84 ±0.23% | 73.77 ±0.44% | 69.91 ±0.49% |
| Direct averaging | 11.40 ±1.62% | 20.55 ±3.07% | 16.90 ±2.02% |
| Permute | 62.11 ±0.30% | 58.45 ±1.76% | 43.82 ±1.31% |
| Weight Matching | 58.18 ±0.68% | 55.87 ±1.80% | 41.15 ±1.45% |
| ZipIt! | 61.41 ±0.51% | 57.97 ±1.29% | **55.08 ±0.70%** |
| **CCA Merge (ours)** | **66.35 ±0.19%** | **60.38 ±1.68%** | 46.57 ±0.76% |

Table 2: ResNet20×8 trained on 3 different splits of CIFAR100 - Accuracies and standard deviations from 4 different merges of 2 models are presented. When the models being merged have learned different features from disjoint sets of the training data but with all the classes (splits 1 and 2) CCA Merge is the only model merging method which outperforms the average of the base models. In the case where the models being merged were trained on disjoint subsets of the classes (split 3) CCA Merge still outperforms past model merging methods except for ZipIt!.

natural in the context of federated or distributed learning. We consider ResNet20 models trained on 3 different data splits of the CIFAR100 training dataset. The first (1) data split is the one considered in Ainsworth et al. (2023) and Jordan et al. (2023) where one model is trained on 80% of the data from the first 50 classes of the CIFAR100 dataset and 20% of the data from the last 50 classes, the second model being trained on the remaining examples. In the second (2) data split we use samples from a Dirichlet distribution with parameter vector $\alpha = (0.5, 0.5)$ to subsamble each class in the dataset and create 2 disjoint data splits, one for each model to be trained on. Lastly, with the third (3) data split we consider the more extreme scenario from Stoica et al. (2024) where one model is trained on 100% of the data from 50 classes, picked at random, and the second one is trained on the remaining classes, with no overlap. For this last setting, in order for both models to have a common output space they were trained using the CLIP Radford et al. (2021) embeddings of the class names as training objectives. In Table 2 we report mean and standard deviation of accuracies across 4 different model pairs.

For the first two data splits CCA Merge outperforms the other methods, beating the second best method by ~4% and ~2% on the first and second data splits respectively. For the third data split CCA Merge is the second best performing method after ZipIt!. However, we note that the comparison with ZipIt! is somewhat unfair, ZipIt! was designed for this specific setting and it allows the merging of features from the same network to reduce redundancies, thus making it more flexible than the other methods which only perform "alignment". In all cases CCA Merge outperforms or is comparable with the base models average indicating that, to some extent, our method successfully combines different learned features from the two models.

## 5    DISCUSSION AND CONCLUSION

Recent model fusion successes exploit inter-model relationships between neurons by modeling them as permutations before combining them. Here, we argue that, while assuming a one-to-one correspondence between neurons yields interesting merging methods, it is rather limited as not all neurons from one network have an exact match with a neuron from another network. Our proposed *CCA Merge* takes the approach of linearly transforming model parameters beyond permutation-based optimization. This added flexibility allows our method to outperform recent competitive baselines when merging pairs of models trained on the same data or on disjoint splits of the data (Tables 1 and 2). Furthermore, when considering the harder task of merging many models, CCA Merge models showcase remarkable accuracy stability as the number of models merged increases, while past methods suffer debilitating accuracy drops. This suggests a path towards achieving *strong linear connectivity* between a set of models, which is hard to do with permutations (Sharma et al., 2024).

ACKNOWLEDGEMENTS

This work was partially funded by NSERC CGS D scholarship [S.H.]; OpenPhilanthropy [E.B.]; FRQNT-NSERC grant 2023-NOVA-329125 [E.B., G.W.]; Canada CIFAR AI Chair, NSF DMS grant 2327211 and NSERC Discovery grant 03267 [G.W.]. This work is also supported by resources from Compute Canada and Calcul Quebec. The content is solely the responsibility of the authors and does not necessarily represent the views of the funding agencies.

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

## A APPENDIX

### A.1 CCA'S FLEXIBILITY ALLOWS IT TO BETTER MODEL RELATIONS BETWEEN NEURONS

We first aim to illustrate the limits of permutation based matching and the flexibility offered by CCA Merge. Suppose we want to merge two models, $\mathcal{A}$ and $\mathcal{B}$, at a specific merging layer, and let $\{z_i^{\mathcal{M}}\}_{i=1}^n$ denote the set of neurons of model $\mathcal{M} \in \{\mathcal{A}, \mathcal{B}\}$ at that layer. Given the activations of the two sets of neurons in response to a set of given inputs, we can compute the correlation matrix $C$ where element $C_{ij}$ is the correlation between neuron $z_i^{\mathcal{A}}$ and $z_j^{\mathcal{B}}$. For each neuron $z_i^{\mathcal{A}}$, for $1 \leq i \leq n$, the distribution of its correlations with all neurons from model $\mathcal{B}$ is of key interest for the problem of model merging. If, as the permutation hypothesis implies, there exists a one-to-one mapping between $\{z_i^{\mathcal{A}}\}_{i=1}^n$ and $\{z_i^{\mathcal{B}}\}_{i=1}^n$, then we would expect to have one highly correlated neuron for each $z_i^{\mathcal{A}}$ – say $z_j^{\mathcal{B}}$ for some $1 \leq j \leq n$ – and all other correlations $C_{ik}$, $k \neq j$, close to zero. On the other hand, if there are multiple neurons from model $\mathcal{B}$ highly correlated with $z_i^{\mathcal{A}}$, this would indicate that the feature learned by $z_i^{\mathcal{A}}$ is distributed across multiple neurons in model $\mathcal{B}$ – a relationship that CCA Merge would capture.

In the left column of Fig. 2, we plot the distributions of the correlations between two ResNet20x8 models (i.e., all the elements from the correlation matrix $C$) for 2 different merging layers. The vast majority of correlations have values around zero, as expected, since each layer learns multiple different features. In the right column of Fig. 2 we use box plots to show the values of the top 5 correlation values across all $\{z_i^{\mathcal{A}}\}_{i=1}^n$. For each neuron $z_i^{\mathcal{A}}$, we select its top $k$-th correlation from $C$ and we plot these values for all neurons $\{z_i^{\mathcal{A}}\}_{i=1}^n$. For example, for $k = 1$, we take the value $\max_{1 \leq j \leq n} C_{ij}$, for $k = 2$ we take the second largest value from the $i$-th row of $C$, and so on. We observe the top correlations values are all relatively high but none of them approaches full correlation (i.e.,

value of one), suggesting that the feature learned by each neuron $z_i^{\mathcal{A}}$ from model $\mathcal{A}$ is distributed across multiple neurons from $\mathcal{B}$ – namely, those having high correlations – as opposed to having a single highly correlated match.

Given the flexibility of CCA Merge, we expect it to better capture these relationships between the neurons of the two networks. We recall that CCA Merge computes a linear transformation $T$ that matches to each neuron $z_i^{\mathcal{A}}$ a linear combination $z_i^{\mathcal{A}} \approx \sum_{j=1}^{n} T_{ij} z_j^{\mathcal{B}}$ of the neurons in $\mathcal{B}$. We expect the distribution of the coefficients (i.e., elements of $T$) to match the distribution of the correlations ($C_{ij}$ elements), indicating the linear transformation found by CCA Merge adequately models the correlations and relationships between the neurons of the two models. For each neuron $z_i^{\mathcal{A}}$, we select its top $k$-th, for $k \in \{1, 2\}$, correlation from the $i$-th row of $C$ and its top $k$-th coefficient from the $i$-th row of $T$ and we plot a histogram of these values for all neurons $\{z_k^{\mathcal{A}}\}_{k=1}^{n}$ in Fig. 3. Indeed, the distributions of the correlations and those of the CCA Merge coefficients are visually similar, albeit not fully coinciding. To quantify this similarity we compute the Wasserstein distance between these distributions, normalized by the equivalent quantity if the transformation were a permutation matrix. For a permutation matrix, the top 1 values would be of 1 for every neuron $z_i^{\mathcal{A}}$ and all other values would be 0. We can see that CCA Merge finds coefficients that closely match the distribution of the correlations, more so than simple permutations, since the ratio of the two distances are 0.15 and 0.04, respectively, for top 1 values in the two considered layers and 0.35 and 0.23 for top 2 values.

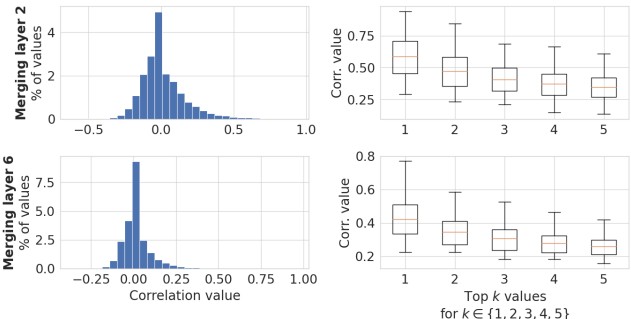

Figure 2: **Left column:** distribution of correlation values between the neurons $\{z_i^{\mathcal{A}}\}_{i=1}^{n}$ and $\{z_i^{\mathcal{B}}\}_{i=1}^{n}$ of two ResNet20x8 models ($\mathcal{A}$ and $\mathcal{B}$) trained on CIFAR100 at two different merging layers; **Right column:** for $k \in \{1, 2, 3, 4, 5\}$ the distributions of the top $k$-th correlation values for all neurons in model $\mathcal{A}$ at those merging layers.

## A.2 EXTENDED RELATED WORK

**Mode connectivity** Freeman & Bruna (2017) proved theoretically that one-layered ReLU neural networks have asymptotically connected level sets, i.e. as the number of neurons increases two minima of such a network are connected by a low loss path in parameter space. Garipov et al. (2018) and Draxler et al. (2018) explore these ideas empirically and introduce the concept of *mode connectivity* to describe ANN minima that are connected by nonlinear paths in parameter space along which the loss remains low, the maximum of the loss along this path was termed the *energy barrier*, and both works proposed algorithms for finding such paths. Garipov et al. (2018) further proposed *Fast Geometric Ensembling* (FGE) as a way to take advantage of mode connectivity by ensembling multiple model checkpoints from a single training trajectory. Frankle et al. (2020) introduced the concept of *linear mode connectivity* which describes the scenario in which two ANN minima are connected by a *linear* low loss path in parameter space. They used linear mode connectivity to study network stability to SGD noise (i.e. different data orders and augmentations). They found that at initialization, networks are typically not stable, i.e. training with different SGD noise from a random initialization typically leads to optima that are not linearly mode connected, however early in training models become stable to such noise.

**Model merging** More recently, Entezari et al. (2022) have conjectured that "Most SGD solutions belong to a set $\mathcal{S}$ whose elements can be permuted in such a way that there is no barrier on the linear

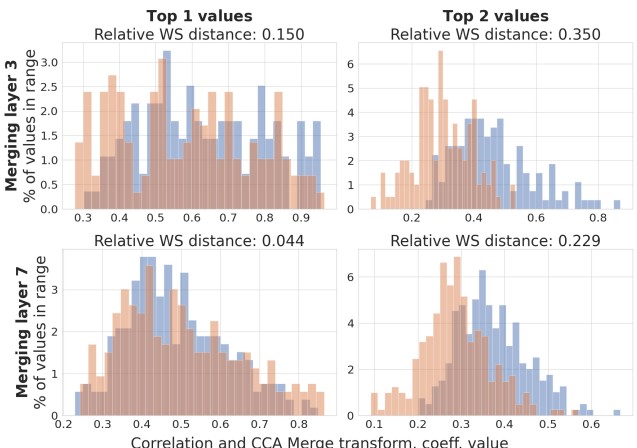

Figure 3: Distributions of top 1 (**left column**) and 2 (**right column**) correlations (blue) and CCA Merge transformation coefficients (orange) across neurons from model $\mathcal{A}$ at two different merging layers. In the left column for example, for each neuron $z_i^{\mathcal{A}}$ we have one correlation value corresponding to $\max_{1 \leq j \leq n} C_{ij}$ and one coefficient value corresponding to $\max_{1 \leq j \leq n} T_{ij}$ where $C$ is the cross-correlation matrix between neurons of models $\mathcal{A}$ and $\mathcal{B}$, and $T$ is the CCA Merge transformation matching neurons of $\mathcal{B}$ to those of $\mathcal{A}$. Wasserstein distance between the distributions of top $k \in \{1, 2\}$ correlations and top $k$ Merge CCA coefficients are reported, relative to equivalent distances between correlations and Permute transforms (all top 1 values are 1, and top 2 values are 0).

interpolation between any two permuted elements in $\mathcal{S}$" or in other words most SGD solutions are linearly mode connected provided the right permutation is applied to align the two solutions. They then perform experiments that support this conjecture and also empirically establish that increasing network width, decreasing depth, using more expressive architectures, or training on a simpler task all help linear mode connectivity by decreasing the loss barrier between two optima. Many other works seem to support this conjecture. For example, Singh & Jaggi (2020) and Peña et al. (2023) propose optimal transport-based methods for finding the best transformation to match two models. Ainsworth et al. (2023) propose an algorithm for finding the optimal permutation for merging models based on the distances between the weights of the models themselves. Jordan et al. (2023) exposes the phenomenon in which interpolated deep networks suffer a variance collapse in their activations leading to poor performance. They propose REPAIR which mitigates this problem by rescaling the preactivations of interpolated networks through the recomputation of BatchNorm statistics.

**"Easy" settings for model averaging** Linear mode connectivity is hard to achieve in deep learning models. Frankle et al. (2020) established that even models being trained on the same dataset with the same learning procedure and even the same initialization might not be linearly mode connected if they have different data orders/augmentations. It seems that only models that are already very close in parameter space can be directly combined through linear interpolation. This is the case for snapshots of a model taken at different points during its training trajectory (Garipov et al., 2018; Izmailov et al., 2018) or multiple fine-tuned models with the same pre-trained initialization (Wortsman et al., 2022; Ilharco et al., 2023; Yadav et al., 2023). This latter setting is the one typically considered in NLP research. Another setting that is worth mentioning here is the "federated learning" inspired one where models are merged every couple of epochs during training (Jolicoeur-Martineau et al., 2023). The common starting point in parameter space and the small number of training iterations before merging make LMC easier to attain.

We emphasize that these settings are different from ours in which we aim to merge *fully trained models* with different parameter initializations and SGD noise (data order and augmentations).

**Merging multiple models** Merging more than two models has only been explored thoroughly in the "easy" settings stated above. For example Wortsman et al. (2022) averages models fine-tuned with different hyperparameter configurations and finds that this improves accuracy and robustness.

Jolicoeur-Martineau et al. (2023) averages the weights of a population of neural networks multiple times during training, leading to performance gains. On the other hand, works that have focused on providing feature alignment methods to be able to merge models in settings in which LMC is not trivial have mainly done so for 2 models at the time (Singh & Jaggi, 2020; Ainsworth et al., 2023; Peña et al., 2023; Jordan et al., 2023). An exception to this is Git Re-Basin (Ainsworth et al., 2023) which proposes a "Merge Many" algorithm for merging a set of multiple models by successively aligning each model to the average of all the other models. However, results obtained with this method, which they use to merge up to 32 models, are relegated to the appendix and only concern the very simple set-up of MLPs on MNIST. Singh & Jaggi (2020) also consider merging multiple models but either in a similarly simple set-up, i.e. 4 MLPs trained on MNIST, or they fine-tune the resulting model after merging up to 8 VGG11 models trained on CIFAR100. We extend this line of work to more challenging settings, using more complex model architectures, we report the merged models accuracies directly without fine-tuning and make this a key focus in our work.

**Model merging beyond permutations**   We note that the two model merging methods based on optimal transport Singh & Jaggi (2020); Peña et al. (2023) can also align models beyond permutations. However, in Singh & Jaggi (2020) this only happens when the two models being merged have different numbers of neurons at each layer. When the models have the same number of neurons the alignment matrix found by their method is a permutation, as such the majority of their results are with permutations. The method proposed by Peña et al. (2023) isn't constrained to finding binary permutation matrices but binarity is still encouraged through the addition of an entropy regularizer. Furthermore, our CCA based method is different in nature from both of these since it is not inspired by optimal transport theory.

**CCA in deep learning**   Canonical Correlation Analysis is a very popular statistical method used in many fields of science De Bie et al. (2005). In the context of deep learning, CCA has been used to align and compare learned representations in deep learning models Raghu et al. (2017); Morcos et al. (2018), a task which is very similar to the feature matching considered by model merging algorithms.

