# OpenReview forum: "Harmony in Diversity: Merging Neural Networks with Canonical Correlation Analysis"
_ICLR.cc/2024/Workshop/Re-Align — ICLR 2024 Workshop Re-Align Poster_

### Official Review · Reviewer_ncFz · 2024-02-22
**Novel alignment method with limited theoretical explanation**

**Rating:** 2
**Fit:** 3
**Confidence:** 2

**Workshop Review:**

This paper proposes a new model merging method based on Canonical Correlation Analysis (CCA Merge). Compared to existing alignment methods, CCA Merge yields better performing merged models for VGG and ResNet architectures trained on CIFAR10, CIFAR100, and ImageNet200. CCA Merge also performs significantly better at aligning multiple models.

**Strengths**
- The writing is very clear. The limitations of ensembling, model fusion, and permutation-based alignment methods are explained well. The use of CCA for alignment is well motivated and novel.
- Appendix A.1 provides good intuitions of why permutation is not sufficient.
- Experiment results are impressive. The proposed method produces merged models with significantly better accuracy than all baselines, especially for the VGG11 model.
- The paper addresses the problem of improving alignment among trained models, which is highly relevant to the community at the workshop. In particular, the experiments on merging multiple models demonstrates that CCA is better than past methods at finding the common features learned by groups of neural networks.

**Weaknesses**
- The methodology section could benefit from more explanations and details about CCA.
- While some intuitions are provided in the appendix, the paper does not give a theoretical justification the advantage of CCA over permutation or guarantees of the accuracy of the merged model.
- The end of the first paragraph in section 4.1 states that the merged models are not expected to outperform the base ones. In this case, why do we need the merged model? Can we just use the best base model instead?

**Reason For Not Giving Higher Score:**

There is limited discussion on theoretical justifications for using CCA for alignment.

**Reason For Not Giving Lower Score:**

The proposed method is novel and performs significantly better at alignment tasks than all baselines.

**Reviewer Domain:**

machine learning

---

### Official Review · Reviewer_3zwF · 2024-02-22
**A good paper, recommended for acceptance**

**Rating:** 2
**Fit:** 3
**Confidence:** 2

**Workshop Review:**

This paper propose a novel method, called CCA Merge, for merging different models. The method is based on the idea of Canonical Correlation Analysis, a statistical method traditionally employed to discern the maximal correlation between linear composites derived from two multivariate datasets. The application of CCA in this context is both novel and reasonable.

The paper is easy to follow and provides good presentations. To my understanding, merging different models on the same task can be a promising way to get better performance. Merging in such a "direct" way can be effective, but meanwhile challenging due to the complexity entailed in feature spaces.

The proposed method is simple, elegant and seems interesting. From the results, it outperforms the current merging methods to a certain extent. Nevertheless, we should also note a saline limitations of these methods, which concerns their inconsistent ability to guarantee performance augmentation after merging. Merged models may be worse compared to the separate unmerged models, which makes the proposed method less useful. Additionally, we encourage the authors can give theoretical backups in their future work.

**Reason For Not Giving Higher Score:**

N/A

**Reason For Not Giving Lower Score:**

A good paper can provide interesting insights in regards to the merging model. Meanwhile, the proposed method is simple and interesting, which can outperform the contemporary methods.

**Reviewer Domain:**

machine learning

---

### Official Review · Reviewer_gea2 · 2024-02-22
**Clear, well-motivated, good experiments. Held back by missing code.**

**Rating:** 2
**Fit:** 3
**Confidence:** 2

**Workshop Review:**

Summary: The paper proposes a new model merging method based on CCA, “CCA Merge”. Contrary to prior methods, the proposed method is more flexible by allowing alignment via linear transformations instead of permutations. CCA Merge outperforms prior methods both when merging two and especially multiple models.

Strengths:
- The objective of the paper is clear and well-motivated.
- The claims regarding performance are well supported. However, I was surprised that the gap to pure permutation is quite low in some cases.
- Comprehensive related work section that makes the paper accessible to a wider audience.

Weaknesses:
- No code is released as far as I see. Releasing the code would make the work more useful to the community and guarantee reproducibility.
- The introduction portraits model merging in context of improving performance similar to ensembles. This is in contrast to the results, where merged models are worse than single models even (Table 1). Do I understand correctly that improving performance would be the end goal, but we are not there yet? I think clarification could help here.

Minor comments and questions:
- The merging transformation $T_i$ stems from the optimal matching between the _activations_, but $T_i$ is then applied on the pre-activations. Would it not make more sense to apply CCA on the outputs of a layer before activation function?
- Please describe the training with CLIP embeddings mentioned in Section 4.3.
- Would you say your approach is connected to model stitching [1]? Training a linear stitching layer between layers at equal depth could maybe give you a transformation similar to the one you get via CCA.

[1] Lenc, Karel, and Andrea Vedaldi. "Understanding image representations by measuring their equivariance and equivalence." Proceedings of the IEEE conference on computer vision and pattern recognition. 2015.

**Reason For Not Giving Higher Score:**

- Impact is limited without code.
- While the paper proposes a novel method, merging (and especially comparing) models with more flexibility than permutation is not completely new.

**Reason For Not Giving Lower Score:**

- The proposed method is novel and has solid evidence.
- Overall high quality paper.

**Reviewer Domain:**

machine learning

---

### Author Response · Authors · 2024-05-10

We thank the reviewers for their thoughtful comments and insightful questions. Their constructive criticism will be taken into consideration to help us improve future iterations of our work.

---

### Decision · Program_Chairs · 2024-03-02

Accept (Poster)